Plant data visualisation using network graphs

Mohamad-Matrol Afrina Adlyna 1
http://orcid.org/0000-0001-7038-6170 Chang Siow-Wee 1
Abu Arpah 1 2 arpah@um.edu.my
1 Institute of Biological Sciences, Faculty of Science, University of Malaya , Kuala Lumpur , Malaysia
2 Centre of Research for Computational Sciences and Informatics for Biology, Bioindustry, Environment, Agriculture and Healthcare, University of Malaya , Kuala Lumpur , Malaysia
Zanin Massimiliano
Electronic publication date: 2018 Aug 31
Publication date: 2018
Volume: 6
Electronic Location ID: e5579
Received 2018 Mar 30; Accepted 2018 Aug 12
Copyright: © 2018 Mohamad-Matrol et al.
Copyright year: 2018
Copyright holder: Mohamad-Matrol et al.
License: This is an open access article distributed under the terms of the Creative Commons Attribution License, which permits unrestricted use, distribution, reproduction and adaptation in any medium and for any purpose provided that it is properly attributed. For attribution, the original author(s), title, publication source (PeerJ) and either DOI or URL of the article must be cited.
License URL: https://creativecommons.org/licenses/by/4.0/

Keywords: Network graph, Data visualisation, Ontology, PlantViz, Plant knowledge

Funding: University of Malaya under UMRG grant RP038B-15AET This project was supported by the University of Malaya under UMRG grant (RP038B-15AET). The funders had no role in study design, data collection and analysis, decision to publish or preparation of the manuscript.

==============================
Background

The amount of plant data such as taxonomical classification, morphological characteristics, ecological attributes and geological distribution in textual and image forms has increased rapidly due to emerging research and technologies. Therefore, it is crucial for experts as well as the public to discern meaningful relationships from this vast amount of data using appropriate methods. The data are often presented in lengthy texts and tables, which make gaining new insights difficult. The study proposes a visual-based representation to display data to users in a meaningful way. This method emphasises the relationships between different data sets.

Method

This study involves four main steps which translate text-based results from Extensible Markup Language (XML) serialisation format into graphs. The four steps include: (1) conversion of ontological dataset as graph model data; (2) query from graph model data; (3) transformation of text-based results in XML serialisation format into a graphical form; and (4) display of results to the user via a graphical user interface (GUI). Ontological data for plants and samples of trees and shrubs were used as the dataset to demonstrate how plant-based data could be integrated into the proposed data visualisation.

Results

A visualisation system named plant visualisation system was developed. This system provides a GUI that enables users to perform the query process, as well as a graphical viewer to display the results of the query in the form of a network graph. The efficiency of the developed visualisation system was measured by performing two types of user evaluations: a usability heuristics evaluation, and a query and visualisation evaluation.

Discussion

The relationships between the data were visualised, enabling the users to easily infer the knowledge and correlations between data. The results from the user evaluation show that the proposed visualisation system is suitable for both expert and novice users, with or without computer skills. This technique demonstrates the practicability of using a computer assisted-tool by providing cognitive analysis for understanding relationships between data. Therefore, the results benefit not only botanists, but also novice users, especially those that are interested to know more about plants.

Introduction

Plants play an important role and benefit all forms of life. In total, 452 vascular plant families and ∼381,910 species have been identified by botanists across the world, as reported by Willis (2017). The biological field is full of diversified data inputs (Marx, 2013). Plant data range from complete genome sequences to geographical information of plant species distribution (Armstead et al., 2009; Hughes, 2006). In addition, advancement in research technology and methodology has caused plant data to increase rapidly. However, shortage of skill and time to analyse these data points continue to be significant obstacles. Relationships between data are often left out, which eventually produce less valuable information. In addition, any constraints in textually inspecting the massive amount of data collected may cause valuable points to be discarded (Keim, 2002). Associating data, that is, related to one another, however, will point out unique points of the data, which will provide new information on various knowledge domains. Hence, it is important for researchers to interpret valuable data and present it in an engaging format so that it is easy for the community to understand. This is because viewing data in plain form such as in texts and tables will not be sufficient to provide clear explanation.

Data visualisation is a comprehensive field involving the crossover between mathematics, computer science, cognitive and perception science and engineering (Telea, 2014). Data visualisation is defined as the representation of data using a visual or artistic approach rather than the traditional reporting method (Yuk & Diamond, 2014). Data visualisation plays an important role in many fields such as business (Tegarden, 1999), geography (Groenendyk, 2013) and biology (Chen et al., 2014; Jensen & Papin, 2014; Sedova, Jaroszewski & Godzik, 2016). Visualised data better conveys the unique properties of the data it represents. In particular, visualising biological data helps researchers to view the data from a different angle, leading to new insights. This is because data shown via graphical representation makes it easier for humans to conduct data analysis, as it provides more cognitive support (Tory & Moller, 2004).

The key to effective data visualisation is the selection of the right type of visualisation to match the type of data used such as charts, network graphs, Sankey diagrams and tree maps. Most data visualisations are interactive, enabling users to manipulate and explore the visualisation instead of perusing only a set of fixed diagrams. For instance, IHME Viz Hub (IHME, 2017) compiles a number of datasets related to health problems around the world in the form of interactive maps. Consensus PathDB (Kamburov et al., 2011) is a database that integrates different types of functional interactions between physical entities in the cell and uses network graphs to show the interactions between proteins. Apart from that, EcoCyc (Keseler et al., 2013) is a comprehensive database outlining the genome sequence of Escherichia coli, where molecular data is displayed in a detailed timeline diagram. In addition, EcoCyc also allows users to customise the display to suit their interests, thus enabling them to move from one region of a genome to another.

Another key to effective data visualisation is the combination of functioning visualisation tools. A good visual library that consists of a set of programming languages helps in the design of any kind of visualisation needed. D3.js (Teller, 2013) is an example of a JavaScript-based library, that is, used for creating dynamic and interactive data visualisations in a web browser. It heavily utilises Cascading Style Sheets, Hypertext Markup Language and Scalable Vector Graphics (SVG) standards, which provide controls over the final result to users. D3.js is applied in visualising different types of data such as the speed and direction of the wind (Cook, 2017), obesity rate of adults (Map, 2017) and property prices (Jackson, 2015).

A few botanic-related visualisation systems have been developed previously based on different motivations. For instance, Gramene (Tello-Ruiz et al., 2016) is an open source database that focuses on genomics in crops and a number of model plant species. This database is curated with controlled vocabularies from a set of ontologies. Besides that, the Botany Array Resource (Toufighi et al., 2005) provides web-based tools of microarray data specifically for plant species and some animal species. Another example is Ensemble Plants (Bolser et al., 2016), which is an online database that contains genome-level information for 39 plant species and functional tools for genomic alignments, functional annotation and other purposes. Apart from that, data visualisation is also used for modelling a virtual three-dimensional plant model in order to simulate the growth process of crops such as the tomato plant modelling system (Lu, Deng & Fei, 2015). However, most available botanical visualisation systems lack interactive elements that allow users to manipulate data, as the systems focus only at the genomic level.

From the studies discussed above, it can be concluded that the huge number of plant data is due to the wide range of plant types and species in existence, together with advancements in research technology. This vast amount of data requires an effective way of presentation for both researchers and members of the community, particularly in a more structured and interactive form. Presenting data in visual form helps to convey a deeper meaning of the data to users, thus encouraging knowledge inference among them. Two important factors that need to be considered in data visualisation are the types of visualisation that best suit the data and the types of visualisation tools for development.

This study looks into how data visualisation can be applied in deducing relationships between plant data, for example, the relationship between taxonomical data, between samples (i.e. physical sample of the plant) and between taxonomical data and samples. Thus, a visual representation for plant data is proposed. The objectives of this study are: (1) to integrate plant-based data, which consists of taxonomical data (in textual format) and samples (in textual format and images); (2) to transform text-based results in Extensible Markup Language (XML) serialisation format into graphical form; and (3) to develop a visualisation system for plant data. In addition, a user evaluation for expert and novice users is conducted to evaluate the proposed visual representation of plant data.

Materials and Methods

Visual-based representation framework

The framework used in this study illustrates the flow of data representation from the database, that is, transformed into visualisation. Figure 1 shows a generic flowchart of the proposed visual representation for plant data. There are four main steps involved: (1) conversion of the ontological dataset into graph data model; (2) query from graph data model; (3) transformation of text-based results in XML serialisation format into graphical form; and (4) display of the result to the user via a graphical user interface (GUI).

Figure 1 A proposed visual-based representation framework.

Generic flowchart of plant data visualisation.

Dataset

The dataset used in this study was obtained from Plant Ontology UM (POUM), an ontology that contains tree and shrub data collected from University of Malaya (UM), Kuala Lumpur, Malaysia. The following steps show the preparation of the dataset used in this study.

Step 1: Define the plant data description

Figure 2 shows the plant data description in this study. Taxonomical classification describes the rank of a plant from species to kingdom level along with the common name, authorship and year. Plant morphological characteristics describe the features of the bark, leaf, fruit, flower and the plant as a whole tree, at the species level. Ecological attributes describe the level of water usage for plant growth, as well as the type of soil and habitat. Geological distribution describes the location and GPS coordinates of the plant sampling area.

Figure 2 Plant data description.

A plant is described with the taxonomical classification, morphological characteristics, ecological attributes, geological distribution and the plant images. Photo credit: the photo archive at UM Plant Knowledge (http://103.18.1.10:8080/plantviz/).

In addition, the plant samples for each species were described and images of the leaf, flower, bark and fruit of the whole tree were collected as well. The morphological characteristics, ecological attributes and geological distribution of plant samples were described individually due to the fact that the difference of habitat environments at plant sampling areas (see Fig. 3) affect the appearances of the plant samples.

Figure 3 Plant sampling areas in UM.

There are five main locations: DTC, Fakulti Kejuruteraan, Fakulti Perniagaan dan Perakaunan, Fakulti Sains and Tasik Varsiti. Map data © 2018 Google.

Step 2: Build the proposed ontology

Plant data description as shown in Fig. 2 was then translated into the proposed ontology schema as shown in Fig. 4. The concepts (the entity), the concepts’ properties (the data property) and the relationships between concepts (the object property) of the domain were determined in this schema.

Figure 4 Proposed ontology schema.

The plant data description translated into ontology in a graph format.

To prevent any misunderstanding in the terms used to describe the plant, a standardised set of vocabularies was designed. These vocabularies were adapted from the existing schema, namely the Biodiversity Information Standards (TDWG, 2018) which consists of a number of specific biodiversity data standards such as Life Sciences Identifier, Darwin Core and TDWG Access Protocol for Information Retrieval. Besides that, few vocabularies were also newly defined because these vocabularies are unavailable in any existing schema.

This ontology schema was then converted into an ontological form, a machine-readable formal specification (in owl file format) which proceeded to the reasoning process to complete the process of plant data annotation. Protégé 5.2 was used for these purposes. Figure 5 shows a partial of POUM using OntoGraph plug-in in Protégé 5.2.

Figure 5 Top-level entities in POUM ontology.

A partial of POUM using OntoGraph plug-in in Protégé 5.2.

There are more than 200 images of plants in the image database. These images are annotated with vocabularies from the POUM, consisting of eight main classes, five sub-classes, 17 object properties and 39 data properties (Table S1). There are 43 species of 42 genera for trees and 31 species of 28 genera for shrubs (Tables S2 and S3) with a total of 222 samples. As shown in Fig. 6, the image of a plant sample is described using sample information such as attributes of the sample’s bark, leaf, flower and fruit, object of the image, location of where the sample was collected, and taxonomic data that focuses on taxonomical ranks and plant characteristics used in describing the plant species.

Figure 6 An example of images and description of Saraca thaipingensis sample.

(A) Digital images representing the S. thaipingensis sample by a whole tree and its parts, namely leaves, bark, flower and fruit. (B) Unique ID and location data of S. thaipingensis sample. (C) Description of characteristics of S. thaipingensis sample. Photo credit: the photo archive at UM Plant Knowledge (http://103.18.1.10:8080/plantviz/).

Workflow

Figure 7 shows the workflow of passing ontological data into the creation of a network graph.

Figure 7 Workflow of network graph development.

Processes involve passing ontological data to the creation of a network graph.

Step 1: Conversion of ontological dataset into a graph data model

Ontological data of the POUM in owl file format is commonly encoded in a Resource Description Framework (RDF) data model with XML syntax. Jena (2017) introduced the serialisation of ontological data into an RDF graph data model before it was queried.

Step 2: Query from graph data model

The query languages used were Simple Protocol and RDF Query Language (SPARQL). RDF query language retrieves and manipulates any data stored in RDF format (Harris, Seaborne & Prud’hommeaux, 2013). Once users submit a text query, an SPARQL query is sent to the server, where querying is executed to the RDF graph data model of the POUM. The results of the query are in the form of XML syntax and then further structured into a JSON format text to be used by the D3.js library.

Step 3: Transformation of text-based results in XML serialisation format into graphical form

The relationship between data is crucial for highlighting data association in the knowledge domain. Three types of relationships are taken into consideration: The relationship between one taxon to another taxon. A taxon is linked to another by its family name. For example, Delonix regia and Acacia auriculiformis are linked to one another as both are in the same family, Fabaceae.

The relationship between a taxa and its sample. Each taxon has three samples and each sample has a unique identifier. For instance, Murraya paniculata has three samples, namely ‘SMurPan001’, ‘SMurPan002’ and ‘SMurPan003’.

The relationship between samples. Samples are related when they are from the same taxon or are obtained from the same location. For instance, samples of Lagerstroemia indica, Manihot esculenta and Terminalia catappa are collected from the same location, which is ‘DTC UM’.

These relationships can be clearly shown when illustrated to users in a visual form. In order to transform texts into a graphical form, the D3.js visualisation library was chosen, as this utilises different types of programming languages in the data visualisation design. The library aims to shorten the development process while maintaining the quality of the system. There are many types of visualisations that can be designed using D3.js; thus, it is important to choose a suitable type of visualisation.

For the type of visualisation in this study, a network graph was chosen. This graph enables the visualisation of plant data, as the relationship between one data to another can be illustrated clearly. The network graph is a type of graph that highlights the relationship between entities and consists of ‘nodes’ as entities and ‘links’ as lines to link between entities. There are two types of nodes, which are the ‘parent’ (PN) and ‘children’ (CN) nodes. PN is a type of node that has one or more CN. Meanwhile, CN is a type of node that has a PN ancestor.

Before generating the graphic representation, a canvas was first set up. The D3.js library uses a layout controlled by force and SVG as the container for the visualisation. A few parameters were defined such as distance, gravity and size from the function d3.layout.force(). The result in JSON format is called a variable, as it was further arranged using D3.js to create a network graph. Besides that, some interactive features such as view node label, highlight node’s links, expand or shrink group of nodes, page of sample information and thumbnail images of plant sample were added into the network graph in order to develop a user-friendly system. This is to enable the interaction between user and data. Hence, users were able to manipulate the content of the network graph with the implemented features.

Step 4: Display result to the user in a graphical user interface

The generated network graph was then displayed in the GUI to provide the full experience to users, as this was embedded in the interface.

Data and source code for the plant visualisation system (PlantViz) development can be accessed and retrieved from https://github.com/afrinaad/PlantViz/.

User testing

There are a variety of data visualisation tools available to help developers achieve the objective of data visualisation. Therefore, it is desirable to determine whether or not the developed visualisation is successful in achieving user needs. In order to measure the performance of the developed plant data visualisation, a user evaluation was carried out. User evaluation consists of two tests, namely: (1) usability heuristics and (2) query and visualisation evaluation. User evaluation involved expert users with research background or experience in the botanical field including users that might have little skill in information technology (IT). Meanwhile, novice users who have little or no research background or experience in the botanical field might have skills in IT.

The usability heuristic evaluation was adapted from Nielsen’s 10 usability heuristics for user interface (UI) design (Nielsen, 1992). These are the general principles for an interactive UI design. In this study, 10 usability features were adapted to match the developed visualisation system. A sample of the questionnaire used in this study is shown in Fig. 8. Users were given five minutes to explore the GUI before the evaluation. This step was performed to observe the users’ first impression of the visualisation system.

Figure 8 Sample of questionnaire for usability heuristics evaluation.

A total of 10 features were adapted from Nielson’s 10 usability heuristics for UI.

A query and visualisation evaluation was conducted to assess the efficiency of the visualisation system in delivering visualised content to users (Amri, Ltifi & Ayed, 2015; Hearst, Laskowski & Silva, 2016). This helps measure the performance of query sending from the GUI to the server and translates the results into a visual format. In addition, this evaluation assists in the observation of the ability of users to interact with data successfully using features of the visualisation system. Figure 9 shows a sample of the questionnaire for this purpose. Users were given guidelines and instructions for using the developed data visualisation system before performing user evaluation. Both the guidelines and instructions were given to four different cases based on specific search parameters, namely scientific name (Case 1), family name (Case 2), location (Case 3) and water usage (Case 4). Query and visualisation were evaluated via a rating of ‘1’ to ‘5’, where ‘1’-Poor, ‘2’-Fair, ‘3’-Average, ‘4’-Good and ‘5’-Excellent. Comments from the users was taken into consideration in order to improve the developed visualisation system.

Figure 9 Sample of questionnaire for query and visualisation evaluation.

The questionnaire consists of five criteria of query evaluation and eight criteria of visualisation evaluation for the visualisation efficiency assessment.

Next, Fisher’s Exact test and t-test were performed to analyse the outcome of the evaluation. Based on the usability heuristic evaluation, Fisher’s Exact was conducted to check whether or not the GUI of the system is dependent on the users’ knowledge in botanic and IT. Meanwhile, based on the query and visualisation evaluation, a t-test was conducted to check whether or not there was any significant difference between the evaluation done by expert and novice users on the visualisation system.

Results

Plant visualisation system

Based on the proposed visual-based representation for plant species data, a prototype of the web-based plant data visualisation system was developed called PlantViz, as shown in Fig. 10. PlantViz consists of a query tool to search for data in POUM and a graphical viewer to display the retrieved results to users. There are four parameters in the Query page, which are based on scientific name, family name, location and water usage. The scientific name and family name parameters are commonly used as search parameters in many public databases (NRCS, 2017; PFAF, 2017; UCONN, 2017). Besides that, the location parameter is for the location where the plant sample was collected. This was chosen as one of the query parameters because the plant samples in this study were mainly collected from various areas in UM. In addition, PlantViz’s target users are members of the university, who are familiar with these locations such as ‘DTC UM’, ‘Tasik Varsiti UM’ and ‘Fakulti Sains UM’, all locations in UM. The water usage parameter defines the level of water needed by a plant species for its growth. Since parameters such as scientific name and family name were used to represent the taxonomical aspect and location was used to represent the geographical information of the plant. The water usage parameter was chosen to represent the morphology attribute of the plant.

Figure 10 GUI of PlantViz.

Main interface of PlantViz that consists of a query tool and a graphical viewer.

The proposed methodology for transforming text-based results into visualisation form was implemented in PlantViz. Figure 11 shows a fragment of the ontological data serialised into an RDF graph data model that contains a collection of RDF nodes attached to each other by annotated relations. This process was done using a ModelFactory class from the Apache Jena library.

Figure 11 Ontological data in the form of a RDF graph data model.

A fragment of the ontological data in RDF serialisation format.

The RDF graph data model was then queried from when the users used the query tool in PlantViz. Textual query from users was sent as a SPARQL query using QueryFactory and QueryExecutionFactory classes from Apache Jena. The result of the query was in XML format, which was then restructured into JSON format. The flexibility of JSON enables users to organise results according to any arrangement, as JSON can contain any number, Unicode character, Boolean operators, array, object or null value (Bassett, 2015). Figure 12 shows an example of the query results in both XML and JSON formats. The example shown is a shortened part of the result to show the difference in terms of structure for the query result before and after being transformed into JSON. Figure 12A illustrates the results generated by Apache Jena, which is in XML format. This was then rearranged into JSON format and later organised into different arrays based on the relationship of the data, as shown in Fig. 12B. For example, general information on the plant such as its common name, water usage and soil type were organised into the same array. Results of the query in JSON format were then passed to the D3.js library to form a network graph as a visualisation of the query result. For instance, the network graph shown in Fig. 13 is the result of querying the Magnolia figo plant species.

Figure 12 An example of query result in XML and JSON formats.

(A) The results generated by Jena is in XML format. (B) The results in JSON format that show the relationship between the data.

Figure 13 PlantViz data visualisation.

An example of visualisation generated in PlantViz using the scientific name parameter, Magnolia figo.

Three types of data relationships, as mentioned earlier, are clearly shown from the results of each chosen query parameter and depicted in Fig. 14. Moreover, unlike plant ontology (PO; Jaiswal et al., 2005), which provides a static graphical view, the graphical viewer in PlantViz provides interactive elements, which allow users to explore the result (Lohmann et al., 2015), as illustrated in Fig. 15. Table 1 lists the description of visualisation features, functions and its conditions.

Figure 14 Examples of data visualisation for the four query parameters.

(A) Query parameter scientific name. (B) Query parameter family name. (C) Query parameter location. (D) Query parameter water usage.

Figure 15 Interactive elements in PlantViz’s graphical viewer.

(A) View node label, (B) highlight node links, (C) expand or shrink group of nodes, (D) sample information page and (E) thumbnail images of plant sample. Photo credit: the photo archive at UM Plant Knowledge (http://103.18.1.10:8080/plantviz/).

Table 1 List of visualisation features in PlantViz.

Visualisation features	Functions	Conditions	
View node label (Fig. 15A)	To show the label of a node	Visible when the cursor is hovered

	
Highlight node’s links (Fig. 15B)	To show relationship between nodes	Visible when the cursor is hovered

Only highlights links connected to the node

	
Expand or shrink group of nodes (Fig. 15C)	To allow users to expand or shrink a group of nodes	When users click on node of ‘parent’ type

Only node of ‘parent’ type can be expanded or shrunk

	
Sample information page (Fig. 15D)	To redirect users to a new page containing information of plant samples	When node with text ‘More detail’ is clicked

	
Thumbnail images of plant samples (Fig. 15E)	To show thumbnail-sized images of plant samples	Visible when the cursor is hovered on nodes with label ‘Sample ID’

	
Note:

The features’ description, function and its conditions.

PlantViz is freely accessible at http://103.18.1.10:8080/plantviz/. The detailed manual on how to use the PlantViz is provided as Supplemental Article S1.

User evaluation

User evaluation was performed on the querying and visualisation of PlantViz. Sixty users including 30 expert users and 30 novice users participated in this user evaluation. The expert users are botanists and researchers in biodiversity with little IT knowledge while the novice users are undergraduate students from UM.

Figure 16 presents an analysis of the usability heuristics evaluation by both expert and novice users. This shows that the majority of users rated ‘Yes’ for most of the features. As shown in Fig. 16A, E6 has the highest number for the rating ‘Yes’ (all 30 expert users voted ‘Yes’), while for novice users, as shown in Fig. 16B, E2, E6 and E9 have the highest number for the rating ‘Yes’ (all 30 novice users voted ‘Yes’). Meanwhile, for both types of users, E1 has the highest number for rating ‘No’ (16 out of 30 expert users and 14 out of 30 novice users voted ‘No’). This is consistent with prototype development. The guidelines for using the system are available with no status for the system being shown.

Figure 16 Analysis of usability heuristics evaluation by expert and novice users.

The bar chart shows the total number of responses that rated ‘Yes’ and ‘No’ for each feature in the usability heuristics evaluation, as per the (A) expert users and (B) novice users.

Fisher’s Exact test was conducted to check whether or not the user experience in using PlantViz is dependent on their expertise level. The null hypothesis H0 is that there is no difference between the usability heuristics evaluation performed between expert users and novice users. The two-tailed probability (p) value of Fisher’s Exact test on usability heuristics evaluation is 0.312, (p < 0.05), which means that there is no significant difference in the usability heuristic evaluation between expert and novice users. This also indicates that PlantViz’s UI is adequate for all types of users regardless of their IT knowledge.

Analyses of query and visualisation evaluation for all four cases by both expert and novice users are shown in Figs. S1 and S2, respectively. Figure 17 shows the analysis of query evaluation by expert and novice users. The total number of responses for each case is plotted against the evaluation rating. The query evaluation by both types of users is shown in Figs. 17A and 17B. These show similar results whereby the majority of expert and novice users gave a ‘4’ or ‘5’ rating for most of the query criteria. There is one response that gave a rating of ‘1’ (one expert user) and a total of nine responses that gave a rating of ‘2’ (six expert and three novice users). A low rating was given for Q4, as shown in Fig. S1.

Figure 17 Analysis of query evaluation by expert and novice users.

The bar chart shows the total number of responses for each rating of all four cases by (A) expert users and (B) novice users.

Furthermore, Fig. 18 shows the analysis of visualisation evaluation by expert and novice users. The total number of responses for each case is plotted against the evaluation rating as well. Figures 18A and 18B show that both types of users rated ‘4’ for most of the visualisation criteria. There is one response with a rating of ‘1’ and 35 responses with a rating of ‘2’ (see Fig. S2). V7 was given a rating of ‘1’ (one expert user rated ‘1’) and had the highest number for a rating of ‘2’ (four experts and nine novice users rated ‘2’). It can therefore be concluded that the PlantViz interactive feature for the sample information page should be improved and be more easily accessible. Besides that, V6 had seven responses with a rating of ‘2’ (two experts and five novice users rated ‘2’), which shows that users did not agree that the information shown in the graphical viewer was sufficient. More information could be added to the network graph such as type of life cycle, propagation method and characteristics of other parts of plants. Linking the data to other external databases such as the accession term in Trait Ontology (Walls et al., 2012) and PO would extend the information scope to users.

Figure 18 Analysis of visualisation evaluation by expert and novice users.

The bar chart shows the total number of responses for each rating for all four cases by (A) expert users and (B) novice users.

Independent sample t-tests were conducted to test the significance of the evaluation done by both types of users. The null hypothesis H0 declares that there is no difference between the evaluation of expert and novice users. T-tests were performed on each case for both evaluations with results as shown in Table 2. The p-values for all cases are higher than the significance level, thus there is a lack of evidence to reject H0. Hence, it can be concluded that there is no statistically significant difference between the query and visualisation evaluation performed by expert and novice users. This signifies that PlantViz can be used by users of all levels, with or without knowledge in the botanical or IT fields.

Table 2 t-test results.

Cases	p-Value	
Query evaluation	Visualisation evaluation	
Case 1	0.082	0.133	
Case 2	0.105	0.165	
Case 3	0.177	0.172	
Case 4	0.225	0.409	
Note:

Results of independent sample t-test for query and visualisation evaluation.

Discussion

Many plant species share resemblance in appearance, yet there are many characteristics such as genomic data, morphological attributes and geographical attributes that can distinguish one plant species from the other. More often than not, the data or information in textual form found on many online databases are presented using lengthy and wordy descriptions. Consequently, this hinders researchers from deducing new knowledge from the presented plant data. In this paper, we report an alternative approach for presenting plant data, which is via network graph. Thus, by visualising the retrieved results from the database, the problem of lengthy retrieved texts as part of the results can be eliminated. In addition, this method emphasises the relationship between data and presents this information in a visualised form for users. Hence, from the visualised data, the relationship between data can be easily analysed and inferred.

As mentioned earlier, PlantViz consists of a query tool and a graphical viewer. Users can use PlantViz via a user-friendly GUI. Thus, two types of tests were conducted to assess the performance efficiency of PlantViz, namely usability heuristics evaluation and query and visualisation evaluations.

From the observation, the p-values of the query and visualisation evaluation (Table 2) share the same pattern, with Case 1 having the lowest p-value. It can be presumed that in Case 1, users were not yet familiar with the graphical viewer of the PlantViz system. This is because the graphical viewer is not a common tool in many public plant-based databases such as The Plant List (List, 2013) and NParks Flora & Fauna (https://florafaunaweb.nparks.gov.sg/Home.aspx). However, all other cases, namely Case 2, Case 3 and Case 4, have p-values higher than Case 1 for both evaluations; hence, the previous assumption is valid as users had just started to become acquainted with PlantViz. In addition, the GUI design of both the query tool and graphical viewer in PlantViz is simple yet still appropriate for both types of users. As for Case 4, both evaluations had the highest p-value. Therefore, it can be concluded that users were easily accustomed to the PlantViz system. This also verifies that the GUI for PlantViz is consistent throughout all cases, whereby each case uses different search parameters.

Yet, filtering feature could be added to the graphical viewer to grant users the ability to filter search results. For instance, the location parameter was used as the search parameter for Case 3 and this could generate a cluttered network graph, as many plant species were collected from the same location. Thus, as mentioned by Cline et al. (2007), users could filter the results by selecting only a certain type as shown in the graphical viewer, and this could help in retrieving the most relevant one. In addition, users should be allowed to filter other types of search parameters such as family name, number of plant sample collected or type of plant.

The query method described here is simple—a typical Boolean search is used. Thus, a more robust and efficient query could be achieved by implementing other query and search methods such as the ranking algorithm (Tran, Tsuji & Masuda, 2009; Zhiguo & Zhengjie, 2010) and natural language query processing (Damljanovic, Agatonovic & Cunningham, 2010; Paredes-Valverde et al., 2015; Tablan, Damljanovic & Bontcheva, 2008; Varga et al., 2014; Wang et al., 2007). It is vital for users to query the system without being attached to a fixed one and to select pre-defined parameters. Based on the low-rated Q4, as shown in Fig. S1, users were dissatisfied with the number of search parameters that can be used to query at a time. This limitation will be enhanced in the future by allowing users to add more than one search parameter to narrow down the retrieved results, so that more relevant results could be retrieved.

The POUM currently consists of plant descriptions and their samples in textual and image forms. PlantViz was able to visualise the POUM dataset by showing the relationships between: (1) ‘taxon-taxon’; (2) ‘taxon-sample’ and (3) ‘sample-sample’. This dataset will be extended to include other plant types along with their descriptions and plant systematics, behaviour, ecology and diversity. Thus, POUM could be linked (Hebeler et al., 2009; Smith et al., 2007) as well to other existing ontologies (Tello-Ruiz et al., 2016). For this reason, PlantViz will be further enhanced to support additional features such as: (1) querying from different datasets at a time and show their relationships and (2) providing data analytics by inferring the visualised relationships.

Furthermore, the visualisation feature will also be improved in order to enhance the usability of the proposed visual-based representation for plant data. Other features such as enabling users to choose the types of visualisation to be generated and choosing a colour scheme to differentiate the relationships between data could be considered as well.

In addition, it is possible to apply the visual-based representation for large dataset such as PlantCLEF (Goeau, Bonnet & Joly, 2017) and National Phenology Database (Marsh, 2017) as these two examples have the database backend and provide the visualisation tool. Definitely, the data will be retrieved; however, the visualised data will be cluttered and interactivity might be inefficient. Hence, an extra task pre-processing the retrieved data such as through dimensionality reduction and data pre-fetch may be needed before the data could be visualised. This pre-processing would retrieve a lesser amount of data, whereby dimensionality reduction helps in reducing the computational load (Kaski & Peltonen, 2011), while data pre-fetch helps in improving the response time (Battle, Chang & Stonebraker, 2016). Besides that, this method could be improved by employing the hierarchical exploration method which is commonly used to visualise large and high-dimensional data as proposed by Lin et al. (2013), Yang, Ward & Rundensteiner (2003) and Zinsmaier et al. (2012). This method is practical because it allows data visualisation in different levels of details (Bikakis, 2018).

Conclusion

The field of biology generates thousands of data every day with the advancements in modern tools and technologies. Hence, it is important to implement a proper methodology for retrieving data and making it accessible to users in an effective way. This study focuses on presenting retrieved data from an ontology to users in a visualisation form. Hence, a visual representation of plant data, PlantViz, was proposed. Data from POUM database was converted as a graph data model and queried using SPARQL. Results from the query were then structured into JSON format before being transformed into a visualisation form and then presented to users in GUI form. User evaluation analysis and results show that PlantViz can be used by users of different levels, either expert users from the botanical field, students or laymen with an interest in botany. PlantViz eliminates lengthy texts of information as results for user queries. In addition, users were able to interact with the data directly, as the visualisation is dynamic. Most importantly, this technique demonstrates the practicability of using computer-assisted tools by providing cognitive analysis to understand the relationship between data. Moreover, this study also facilitates users in inferring and gaining new data insights.

Supplemental Information

Supplemental Information 1 Article S1. How-To of PlantViz.

The detailed manual on how to use the PlantViz.

Click here for additional data file.

Supplemental Information 2 Table S1. Classes and properties of POUM.

Standard vocabularies representing the classes and properties used in POUM.

Click here for additional data file.

Supplemental Information 3 Table S2. List of family, genus, and species for tree.

There are 43 species of 42 genera for trees in POUM.

Click here for additional data file.

Supplemental Information 4 Table S3. List of family, genus, and species for shrub.

There are 31 species of 28 genera for shrubs in POUM.

Click here for additional data file.

Supplemental Information 5 Fig. S1. Analyses of query evaluation by both expert and novice users.

Complete analyses of query evaluation for all four cases.

Click here for additional data file.

Supplemental Information 6 Fig. S2. Analyses of visualisation evaluation by both expert and novice users.

Complete analyses of visualisation evaluation for all four cases.

Click here for additional data file.

Additional Information and Declarations

Competing Interests

Author Contributions

Data Availability

The authors declare that they have no competing interests.

Afrina Adlyna Mohamad-Matrol conceived and designed the experiments, performed the experiments, analysed the data, prepared figures and/or tables, authored or reviewed drafts of the paper.

Siow-Wee Chang conceived and designed the experiments, analysed the data, authored or reviewed drafts of the paper.

Arpah Abu conceived and designed the experiments, performed the experiments, analysed the data, prepared figures and/or tables, authored or reviewed drafts of the paper, approved the final draft.

The following information was supplied regarding data availability:

GitHub: https://github.com/afrinaad/PlantViz/

PlantViz is freely accessible at http://103.18.1.10:8080/plantviz/.

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
