# Peer review of "Plant data visualisation using network graphs"

_PeerJ, doi:10.7717/peerj.5579_

## Round 0.1 · original submission · Major Revisions

Dear authors,

As you can see from the reports below, the two Referees have found a set of shortcomings that should be solved before publication. Beyond that, there are a couple of points that I want to clarify.

First of all, the second Referee suggests that this paper may be a case study. I have double checked this with the editorial staff, and the conclusion is that this is a ‘bioinformatics tool’ article. Thus, that comment does not apply.

Yet, I have a comment myself. It is clear that the described system could in principle be used for the analysis of any plant data set (and beyond plants). Yet, there is no explanation on how to prepare new data, in order to be fed into the system; thus, applying the system to new data would require substantial software expertise… From the point of view of a standard user, this system can only analyse the data initially provided.
Therefore, I think you need to clearly explain how new data have to be prepared, in order to be analysed by the system. This would especially be good for the paper, as then the paper would be much more general, and thus would attract more citations.

Finally, I agree with the first Referee’s comment: (1) Where is a demo site? It would be awesome if a demo site could be provided… Especially because then the interested reader could test the system before having to download and install it. Would it be feasible?

Reviewer 1 ·

Basic reporting

well-written.

Experimental design

no comment

Validity of the findings

no comment

Additional comments

The manuscript entitled “Plant data visualization using network graph” by Mohamad-Matrol proposes a web-based visualization system, called PlantViz, using a graph model representation of plant data. The system can convert ontological dataset as graph model, query from the model data, transform lengthy text data form into graphical form, and display the results based on GUI. The authors assessed the PlantViz by two different types of user evaluations, the usability heuristics- and query and visualization evaluations. The results suggested that the PlantViz can contribute to data retrieval and sharing to infer and gain new insights from the vast amount of plant data. Given that biological data integration using Semantic Web technologies like RDF and SPARQL is one of promising approaches, I think that this work is timely and effective. Although the development of the methodology and evaluation seems to be reasonable, I have a few comments below.

(1) Where is a demo site?
As a whole, the manuscript is well-written. The figures and tables are nicely organized. However, I was not able to find the URL link to the demo site on the WWW, although the source code for the PlantViz development can be accessed in GitHub. I am not sure that the system works well.

(2) Detailed manual ‘how-to’
The authors should prepare the detailed tutorial section to include actual "How-to's" on the website and Supplementary materials on the journal site. I also think that multiple example datasets are very useful for potential users and readers in the journal.

·

Basic reporting

Although the paper is on the whole well written, it seems to me that there is a lot of redundant material, and it takes the authors a long time to get to the point (i.e. that they are described a web-based database graphical querying service for a relatively small dataset).

L14 There are some basic errors of grammar (e.g. ‘graph’ should be ‘graphs’ in the title), but there are also some phrases that are ambiguous to the reader, e.g. L14 in the abstract, what type of “botanical data” are the authors referring to? Survey data? Morphological data? Genetic data? This should be clear from the abstract, even on reading the entire abstract it is unclear to me exactly what types of data are going to be dealt with in this paper. (Indeed, this is not broached until L123 of the paper)
L20 of the Methods in the abstract. Again, this is ambiguous, what type of “text-based query results” are the authors referring to? SQL queries of a relational database? API queries to a webservice of some sort? Some other GUI query to a custom catalogue of some type?
L42, ref should be in the format Willis (2007), not (Willis 2007).
L43-49 This very general discussion of woody plants does not seem hugely relevant to the paper.
L62-L122 This seems to contain a large amount of background material, most of which seems tangential to the method being presented in this paper. Do we really need to know about two separate JavaScript libraries? The topic (introduced in L123) should be introduced a lot earlier.

Experimental design

I'm afraid that, to me, this paper seems to be outside of the PeerJ aims and scope. It seems to be describing a fairly special case of constructing a graphical web-based querying service for a small local dataset. The general way in which the authors have approached this challenge may be of interest to a small technical readership interested in doing something similar, but it does not seem to have importance as a piece of research filling a knowledge gap.

L239-241 The descriptions of the two tests here seem to me to be describing the same thing! Whilst I can understand how Fisher's Exact test might be applied to a 5 x 2 table of counts (five levels of usability x two user types for each case), I struggle to understand how this type of data is suitable for a t-test (and how the question being answered by the t-test approach is any different to that being examined by Fisher's Exact text (i.e. did novice and expert users report different usabilities), but perhaps this is just down to a poor explanation of what was actually done). [ In fact, in the results it emerges that the FE test was applied to the usability heuristics data, and the t-test to the other data; this is not at all clear from the methods. I still seriously doubt the appropriateness of a t-test for examining differences in small (20 v. 30) groups of numbers between 1-5 see e.g. https://en.wikipedia.org/wiki/Student%27s_t-test#Assumptions ]

Validity of the findings

As noted above, I don't think this paper meets the PeerJ aims and scope; although, having said that, with the exception of the issues above under (1) Basic reporting and (2) Experimental design (particularly the t-test), the paper could be said to be 'valid', in that it presents a database structure and then asks some people (novices v experts) to test its usability. However, I would classify this firmly as a case study or case report rather than a Research Article (see https://peerj.com/about/aims-and-scope/)

---

## Round 0.2 · Minor Revisions

The authors have undoubtedly improved their manuscript, and the inclusion of the link to the public version of the system is indeed a plus. I would just recommend the authors to double check the English, and include the suggestion of the Referee on the "critical reflection on the wider use of the approach". Then we can indeed publish the work.

·

Basic reporting

In general the revised paper is well written. Although all parts of the manuscript are intelligible to an intelligent native speaker of English, the general standard of English could be improved upon by a copy-editor or proof-reader. All other basic reporting criteria appear to be met.

Experimental design

The research question and methods are now much better described than previously; the comprehensiveness of the figures is particularly good. The description of the statistical tests are also improved. Although I still have my doubts that a t-test is really suitable for the data presented under the query and visualisation evaluation (a point not clearly responded to by the authors in their rebuttal letter), the figures presented make it blindingly clear that there is almost no difference in the evaluations of novices and experts, and so the statistics are barely necessary.

Validity of the findings

The findings are clearly stated, although it would be nice to have a bit of critical reflection on the wider use of the approach by the authors. For example, where countries have millions of sample points for species (e.g. in the UK the national set of species occurrence data held by the database of the Botanical Society of Britain and Ireland totals ~40 million 'samples'), is a visual graph approach likely to be practical?

Additional comments

L120 what type of sample are you referring to? Not quite clear here whether a sample is a record of a species’ occurrence in space and time, or a herbarium or other physical sample of the plant.

---

## Round 0.3 · accepted · Accept

I think all open issues have been successfully tackled.

#